# Development and Application of a QGIS-Based Model to Estimate Monthly Streamflow

Hanyong Lee [1], Min Suh Chae [1], Jong-Yoon Park [2], Kyoung Jae Lim [3] and Youn Shik Park [1,*]

1   Department of Rural Construction Engineering, Kongju National University, 54 Daehak-ro,
    Yesan-gun 32439, Korea; hylee@smail.kongju.ac.kr (H.L.); cswoo6432@smail.kongju.ac.kr (M.S.C.)
2   Korea Environment Institute, 370 Sicheong-daero, Sejong 30147, Korea; jongyoonpark@kei.re.kr
3   Department of Regional Infrastructures Engineering, Kangwon National University, 1 Gangwondaehakjil,
    Chunsheon-si 24341, Korea; kjlim@kangwon.ac.kr
*   Correspondence: park397@kongju.ac.kr; Tel.: +82-41-330-1267

**Abstract:** Changes in rainfall pattern and land use have caused considerable impacts on the hydrological behavior of watersheds; a Long-Term Hydrologic Impact Analysis (L-THIA) model has been used to simulate such variations. The L-THIA model defines curve number according to the land use and hydrological soil group before calculating the direct runoff based on the amount of rainfall, making it a convenient method of analysis. Recently, a method was proposed to estimate baseflow using this model, which may be used to estimate the overall streamflow. Given that this model considers the spatial distribution of land use and hydrological soil groups and must use rainfall data at multiple positions, it requires the usage of a geographical information system (GIS). Therefore, a model that estimates streamflow using land use maps, hydrologic soil group maps, and rain gauge station maps in QGIS, a popular GIS software, was developed. This model was tested in 15 watersheds.

**Keywords:** GIS-based model; hydrologic watershed modeling; streamflow estimation

## 1. Introduction

The hydrological behavior of a watershed may vary according to changes in rainfall pattern or land use typically caused by urbanization or industrialization [1,2]. These changes may increase the impermeability in the watershed or alter the frequency or amount of rainfall, which decreases the groundwater recharge; thereby, increasing or changing direct runoff, peak runoff, the potential of downstream flooding, and seasonal variance in hydrological behavior [3]. Moreover, it is difficult to secure water resources as soil water retention decreases in a watershed. In addition, due to a possible increase in nonpoint pollution source (NPS) loads from increased direct runoff from streams, watershed management is necessary to secure water resources and control sources of pollution. During watershed management, rainfall patterns or land use change analyses are performed using hydrological models; the Long-Term Hydrologic Impact Analysis (L-THIA) model has been regularly used since its development in 1994 [4–7].

The L-THIA model was initially developed in a spreadsheet format [4], which was later developed for use in geographic information systems (GISs) [5,6]. An ArcView software-based L-THIA/NPS WWW model was developed in 1999 [7], which reflected various land use conditions. Bhaduri et al. [5] and Lim et al. [7] used the L-THIA/NPS WWW model to analyze direct runoff and nonpoint pollution source load according to changes in land use. During the simulation period from 1973 to 1991, an 18% increase in the impermeable area resulted in an 80% increase in the average annual direct runoff and a 50% increase in the nonpoint pollution source load. Tang et al. [8] applied the L-THIA model from 1973–1997 in Little Eagle Creek, Indiana, United States, to analyze the urban design scenario in the area. Results showed that when the urban area increased by 14% from 1973 to 1983, direct runoff increased by 44%; a 34% increase in the urban area

from 1991 to 1997 resulted in only an 11% increase in direct runoff, and it was considered that an appropriate urban design could minimize the variation of direct runoff. Wilson and Weng [9] used the L-THIA model to analyze variations in direct runoff according to changes in land use. When rainfall increased by approximately 30% and the residential area increased by approximately 37%, direct runoff increased by 200% or above. Eaton [10] performed a green infrastructure screening analysis and demonstrated that direct runoff could be reduced by up to 12% by installing a bioretention system and a rain garden in the watershed. Li et al. [11] employed an ArcL-THIA model [12] to analyze the impact of variations in rainfall patterns and land use within the watershed while supplementing the ArcL-THIA model with a baseflow filter program (BFLOW) [13] to separate direct runoff from streamflow; Li et al. [11] stated that variations in rainfall patterns have a higher impact on the direct runoff as compared to changes in land use. Ahiablame et al. [14] assessed low-impact development practices using the L-THIA-LID model, where the impact on baseflow as well as direct runoff was considered in the analysis. Clearly, the L-THIA model has been used to analyze hydrological behavior according to changes in rainfall or land use conditions within various watersheds, but only an estimation of the direct runoff is possible by using the National Resources Conversion Service (formerly Soil Conservation Service) Curve Number (NRCS-CN) method. Moreover, it is difficult to measure direct runoff in watersheds during rainfall. Streamflow is measured as the sum of direct runoff with the baseflow, and it is difficult to conduct an immediate comparison between the results of the estimation and actual measurements. Thus, an exclusive assessment on the direct runoff generally follows the L-THIA model [5,8–10] or another model to separate the direct runoff for comparison in the measured watershed [11]; there were limitations to considering seasonal streamflow variations only from yearly baseflow estimations, despite considering the baseflow parameter [15]. In other words, compared to other hydrological models [16–20], the L-THIA model provides benefits in model simplicity, as the model requires only daily rainfall data, land map, and hydrological soil map, and can be calibrated by only CNs. This is why the model is used and modified consistently. However, the model still has limitations that are challenging to compare model outputs to measured streamflow, and that seasonal variance of flow cannot be estimated.

This study developed an L-THIA model that estimates seasonal variations in streamflow using a monthly baseflow estimation method [21], which can be applied to the recently proposed L-THIA model using land use maps, hydrological soil group maps, and rainfall data measured from multiple sources.

## 2. Materials and Methods

### 2.1. Direct Runoff Calculation

The L-THIA model simulates direct runoff with the NRCS-CN method [22], which uses rainfall (P, mm), initial abstraction ($I_a$, mm), potential maximum retention after runoff begins (S, mm), and the curve number (CN), defined by land use and the hydrological soil group (HSG) to estimate direct runoff depth (Q, mm) (Equations (1)–(4)).

$$Q = \frac{(P - I_a)^2}{(P - I_a) + S} \text{ for } I_a < P \tag{1}$$

$$Q = 0 \text{ for } I_a \geq P \tag{2}$$

$$I_a = 0.2S \tag{3}$$

$$S = \frac{25400}{CN} - 254 \tag{4}$$

Direct runoff depth (Q) was calculated based on CN, which is defined by land use and HSG. Q is thus calculated for every hydrologic response unit (HRU) as a combination of land use and HSG; CN must be determined before calculating Q. S is subsequently calculated using Equation (4), which follows the calculation of $I_a$ from Equation (3). Next, P and $I_a$ are

compared to selectively calculate the Q with either Equation (1) or (2), whichever agrees with the conditions. The option between Equations (1) and (2) allows the consideration of the case of having no direct runoff when rainfall is retained in the soil surface. This retention may vary depending on the amount of water retained in the soil from previous rainfall, which indicates varying amounts of retention despite an equal amount of rainfall. Thus, the soil moisture condition must be reflected in the calculation of Q, which is possible using the antecedent moisture condition (AMC). According to USDA [22], AMC I refers to dry soil conditions (i.e., the soil moisture content is very low/at a wilting point), AMC II refers to a condition where the soil is not overly dry or moist, and AMC III refers to the wet soil condition (i.e., the soil is saturated); AMC is defined by the 5-day antecedent rainfall ($P_5$) and the season (Table 1).

**Table 1.** Definition of AMC.

| AMC | Description | $P_5$ | |
|---|---|---|---|
| | | **Growing Season** | **Dormant Season** |
| AMC I | Dry soil | $P_5 < 35$ mm | $P_5 < 12$ mm |
| AMC II | | $35$ mm $\leq P_5 \leq 53$ mm | $12$ mm $\leq P_5 \leq 28$ mm |
| AMC III | Wet soil | $53$ mm $< P_5$ | $28$ mm $< P_5$ |

Once AMC is determined, the CN must be adjusted accordingly. First, the CN of AMC II is denoted as CN II, which is a value determined by land use and HSG. The CN I for AMG I is determined by Equation (5), and CN II and CN III are determined using Equation (6) and CN II, respectively. Finally, either CN I, CN II, or CN III are applied to Equations (1)–(4) to calculate Q.

$$\text{CN I} = \text{CN II}/(2.281 - 0.0128 \times \text{CN II}) \tag{5}$$

$$\text{CN III} = \text{CN II}/(0.427 - 0.0057 \times \text{CN II}) \tag{6}$$

*2.2. Baseflow Calculation*

The L-THIA model was developed in 1994 [4] and has been since improved to be compatible with the ArcGIS software [12]. The sustained use of this model, mainly in the simulations for direct runoff according to land use and rainfall pattern variations, is because of the simplicity of the model; only input data for determining the CN are required, and the calculation process is simple. Lee et al. [21] stated that the actual streamflow and estimated results from this L-THIA model must be comparable to improve the applicability of the model. This simplicity must be retained throughout the improvement process. Twenty watersheds ranging from 5695 ha to 155,806 ha were thus selected, and the rainfall ($P_i$, mm) of the $i^{th}$ month, total urban area ($A_{URBN}$, ha), area under agriculture ($A_{AGRL}$, ha), area of forest cover ($A_{FRST}$, ha), area used for pasture ($A_{PAST}$, ha), area of wetland cover ($A_{WTDL}$, ha), area of bare land ($A_{BARE}$, ha), water area ($A_{WATR}$, ha), and coefficients for each land use area were used to propose a method for estimating the baseflow ($m^3$) of the $i^{th}$ month (Equation (7)) [21].

$$\text{Baseflow}_i = P_i \times (C_{URBN} \times A_{URBN} + C_{AGRL} \times A_{AGRL} + C_{FRST} \times A_{FRST} + C_{PAST} \times A_{PAST} + C_{WTLD} \times A_{WTLD} + C_{BARE} \times A_{BARE} + C_{WATR} \times A_{WATR}) \times 10.0 \tag{7}$$

In Equation (7), the coefficients for each land use are model parameters that can be calibrated according to the target watershed being used for simulation. Lee et al. [13] proposed the basic values as 0.04 ($C_{URBM}$), 0.40 ($C_{AGRL}$), 0.20 ($C_{FRST}$), 0.18 ($C_{PAST}$), 0.48 ($C_{WTLD}$), 0.15 ($C_{BARE}$), and 0.22 ($C_{WATR}$).

This study used the aforementioned approach to improve the L-THIA model. Area data from each land use area were set to be extracted from the land use map required

for calculating the direct runoff, and no further input data were required for monthly baseflow estimation.

### 2.3. Development of L-THIA 2022 Model

The NRCS-CN method, used for the L-THIA model to estimate the direct runoff, calculates CN with land use and HSG data, estimating direct runoff with this CN value and the rainfall data. However, numerous HRUs are generated by the combination of land use and HSG in the watershed, and there is a realistic limit to the manual calculation of direct runoff for each HRU. Thus, the L-THIA model in this study was developed to be operable in the QGIS software version 3.10 [23].

The calculation process for monthly streamflow involved two steps—direct flow and baseflow calculations. First, the direct flow calculation requires a CN table file that defines the CN values according to the land use and HSG, and a HSG map which carries soil spatial data. The CN table file in a comma-separated value (CSV) file format can be calibrated by the user; the file is provided along with the L-THIA 2022 model. Thus, after overlapping the land use and HSG maps, the CN values are obtained from the CN table file to generate a CN map (Figure 1). In the case of having multiple rainfall measurement points around the watershed in which the L-THIA 2022 is applied, rainfall data measured from these points must be considered. This model was developed to receive the rain gauge station and subwatershed maps and to apply rainfall data from the nearest rain gauge station from each subwatershed. After this process, an HRU information file is generated, which holds the CN, area, and rain gauge station data for all HRUs. The daily direct runoff was estimated based on the NRCS-CN method at each HRU, by using the HRU information and rainfall data files, based on rainfall data from the nearest rain gauge station among the given multiple stations. The sum of daily direct runoff for all HRUs is then converted to the monthly direct runoff for each subwatershed (Figure 1).

The land use maps, subwatershed maps, rain gauge station maps, and rainfall data files are necessary for calculating baseflow. First, land use and subwatershed maps are used to draw land use data for each subwatershed, and the rain gauge station applied to each subwatershed is defined using the subwatershed and rain gauge station maps. The land use area for each subwatershed and the monthly rainfall are applied to Equation (7) to estimate the monthly baseflow for each subwatershed (Figure 1).

The L-THIA 2022 model was developed to be operable in QGIS 3.10, and it is composed of four interfaces (Figure 2). The first interface generates the CN map necessary for estimating direct runoff and requires land use and HSG maps as input data (Figure 2a). In the second interface, the subwatershed map, gauge station map, and daily rainfall data files are required as input data, which is then used to estimate the daily direct runoff (Figure 2c). The CN file in the CSV format can be adjusted in the Microsoft Excel software and is matched with the CN map created from the first interface to define the spatially distributed CN of the HRU. Furthermore, direct runoff is calculated with this spatially distributed CN of the HRU, for which the CNs require adjusting during model calibration. Microsoft Excel must be used to edit CN files, assisting the cumbersome processing of the CN map; this must be repeated multiple times by comparing data with the measured streamflow (or direct runoff) until a satisfactory estimated streamflow (or direct runoff) is achieved. Therefore, the model in the study was developed to enable the application of the immediately adjusted CN in the model interface (Figure 2c). The fourth interface is used for estimating the baseflow of each subwatershed and for classifying the user's land use according to the land use classification (i.e., urban agriculture, forest, pasture, wetland, bare land, or water) to estimate the monthly baseflow, as seen in Equation (7). In addition, the coefficients for each land use in Equation (7) can be adjusted in this interface (Figure 2d).

### 2.4. Application of L-THIA 2022

This study proposed an L-THIA 2022 model, which is operable in the QGIS software, for estimating monthly streamflow. In addition, a test application of the L-THIA 2022

model was performed, for which the land use map data (scale at 1:5000) were provided by the Environmental Geographic Information Service [24], the monthly rainfall provided by the Korea Meteorological Administration (KMA) [25], and the daily flow rate provided by the Water Resources Management Information System (WAMIS) [26]. In the watershed for test application, flow measurement points from WAMIS were set as watershed outlets and selected to avoid any spatial overlap. A total of 15 watersheds were selected (Figure 3), and the watershed areas ranged from a minimum of 5841 to a maximum of 81,107 ha (Table 2). The land use of each watershed was classified into urban, agriculture, forest, pasture, wetland, bare land, and water. Forests accounted for the largest area in each watershed excluding Wsd-02, followed by agriculture.

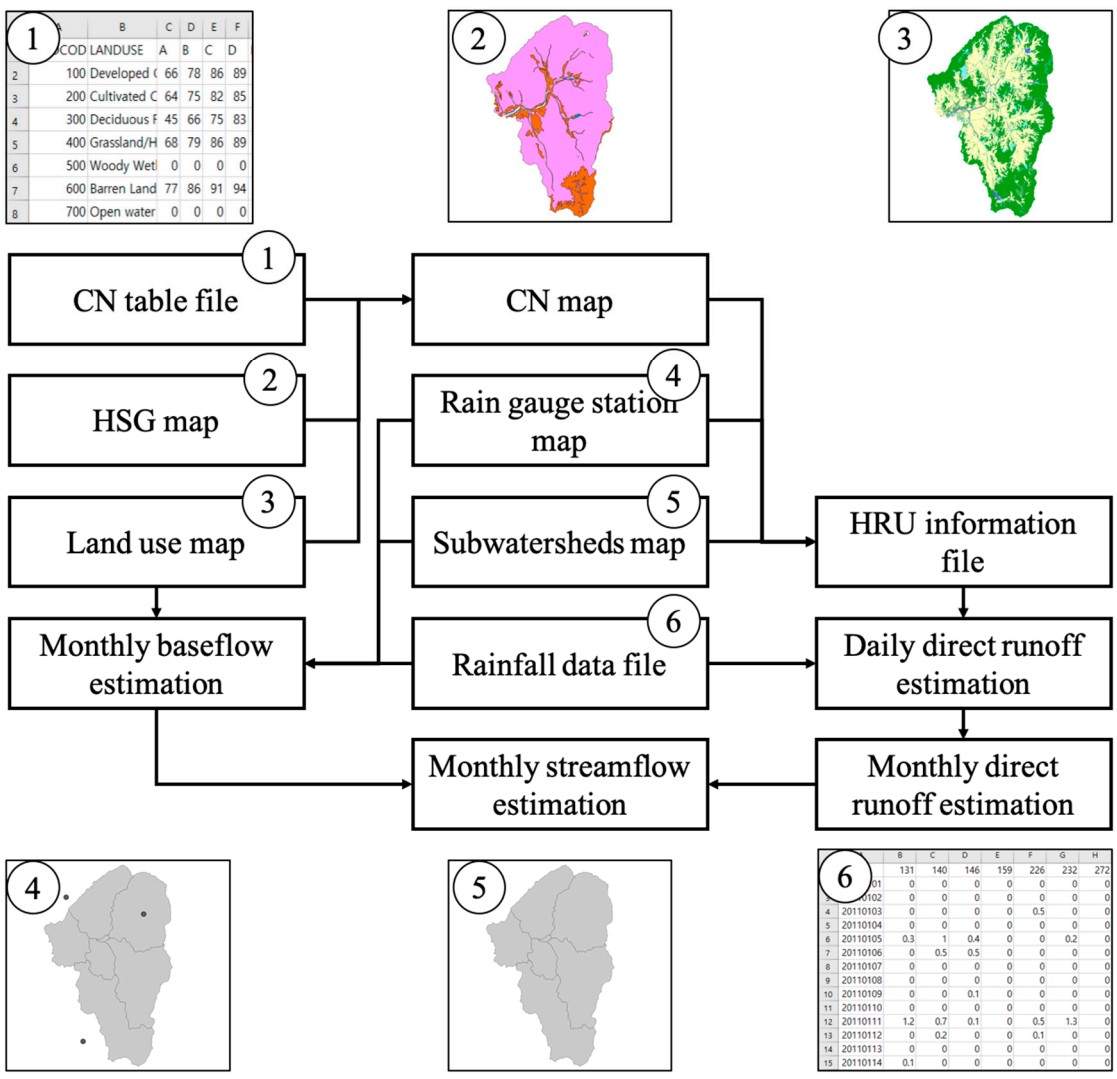

**Figure 1.** Schematic depicting the L-THIA 2022 model.

The test application period was from January 2011 to December 2020, and the minimum and maximum monthly streamflows in each watershed were 0.10 m$^3$ in Wsd-09 and 462.88 m$^3$ in Wsd-15, respectively. The L-THIA model uses rainfall data collected from numerous rain gauge stations located near or within watersheds. Therefore, rainfall data collected from the greatest possible number of rain gauge stations were used, for which measurement data from at least one rain gauge station in each of six watersheds, including Wsd-02 and measurement data from up to four rain gauge stations in the Wsd-09 watershed, were used (Table 3).

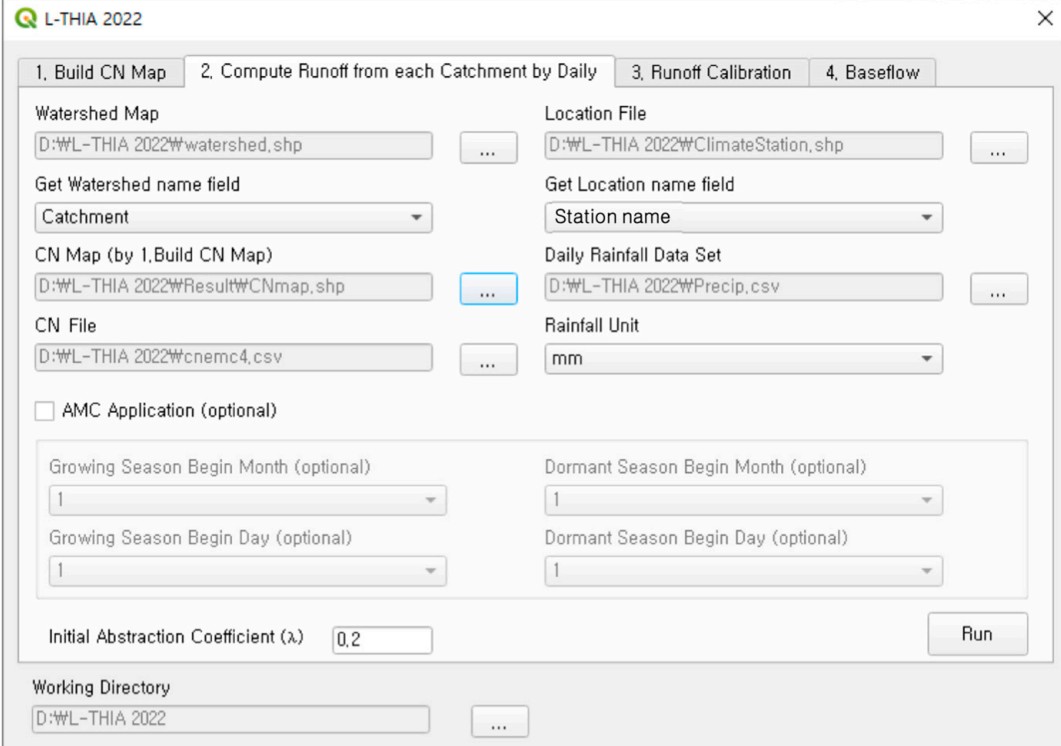

(**a**)

(**b**)

**Figure 2.** *Cont.*

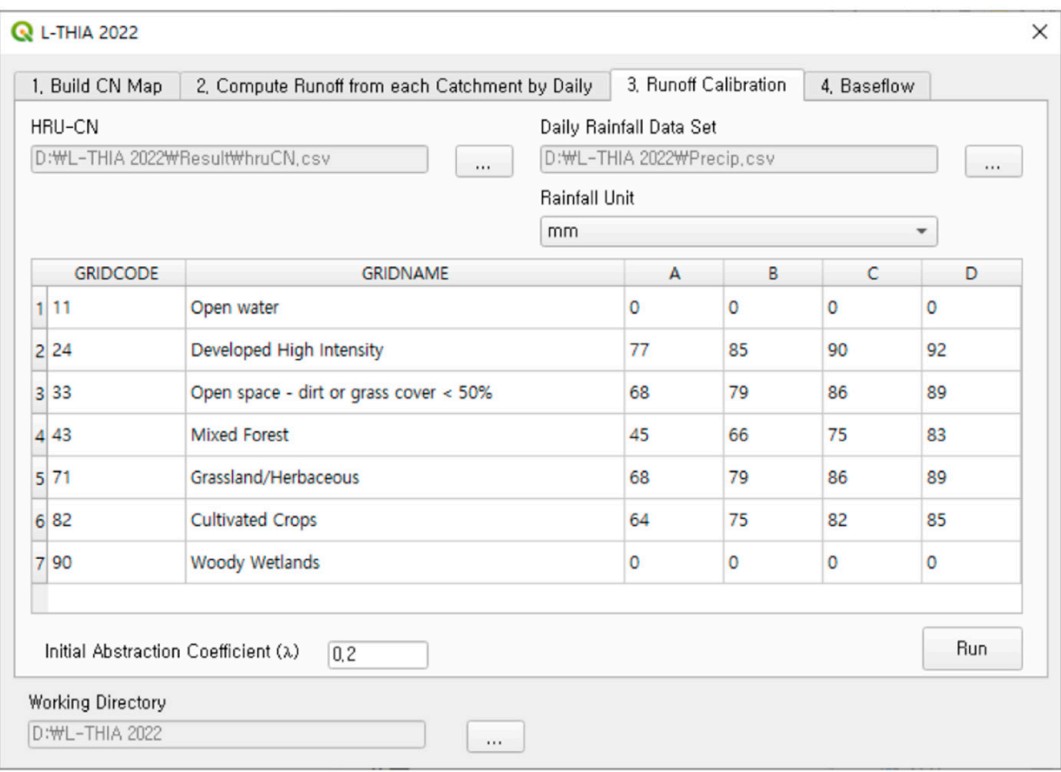

(c)

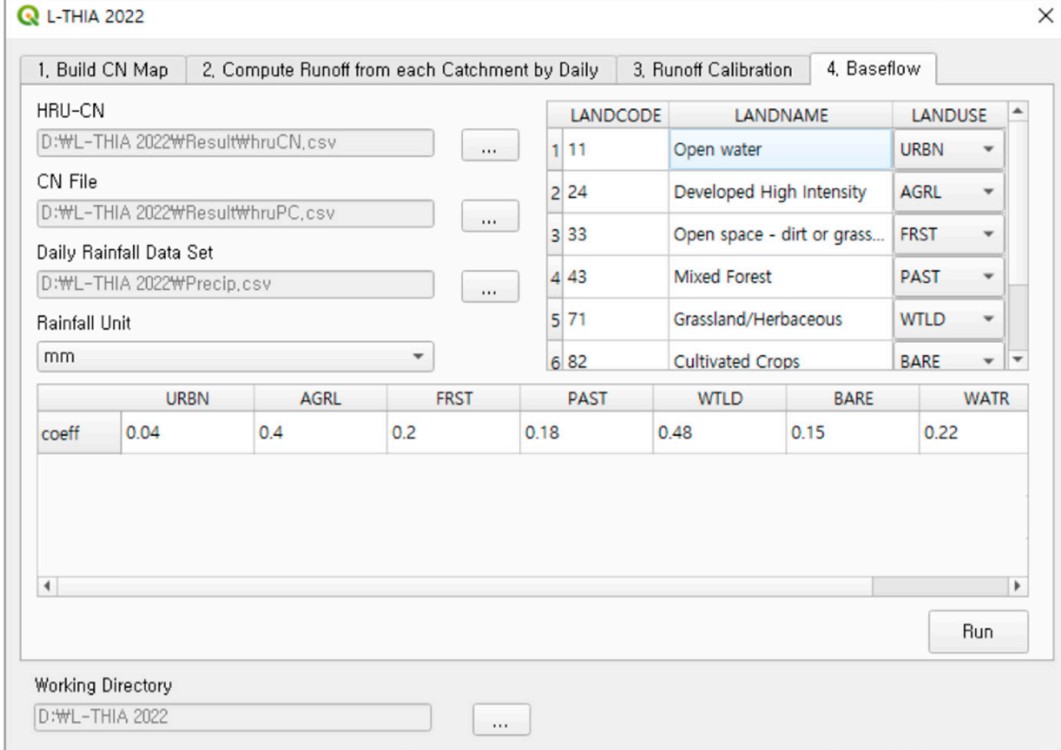

(d)

**Figure 2.** Interfaces of the L-THIA 2022 model: (**a**) an interface to build a CN map, (**b**) an interface to compute daily direct runoff, (**c**) an interface to update CNs, and (**d**) an interface to compute monthly baseflow.

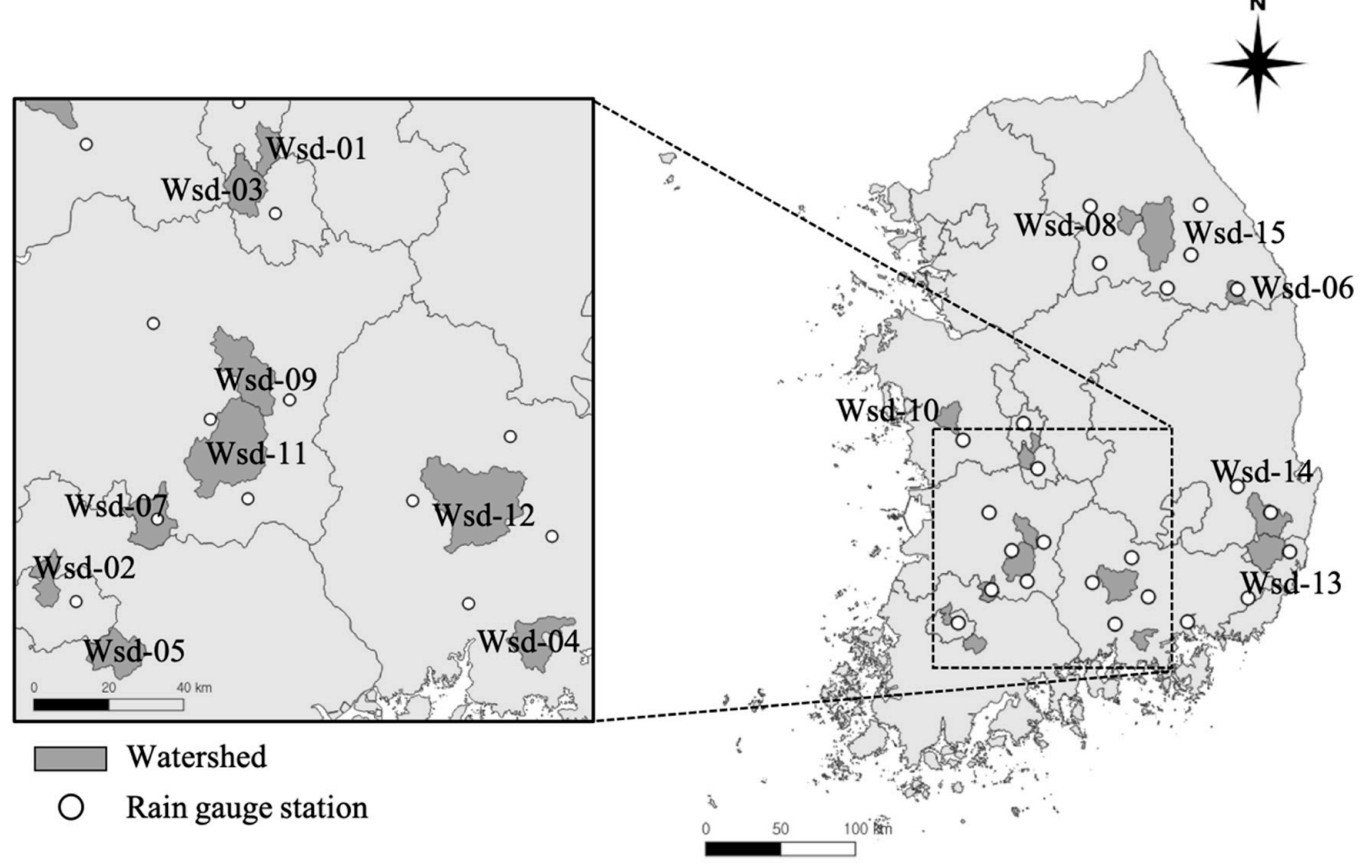

**Figure 3.** Locations of studied watersheds and rain gauge stations.

**Table 2.** Land use in watersheds.

| Watershed | Area (ha) | | | | | | | |
|---|---|---|---|---|---|---|---|---|
| | Urban | Agriculture | Forest | Pasture | Wetland | Bare land | Water | Total |
| Wsd-01 | 657 | 493 | 3931 | 568 | 50 | 123 | 18 | 5841 |
| Wsd-02 | 2022 | 2473 | 1067 | 1035 | 61 | 270 | 82 | 7012 |
| Wsd-03 | 475 | 1204 | 8204 | 1474 | 110 | 286 | 68 | 11,821 |
| Wsd-04 | 420 | 2263 | 8631 | 788 | 197 | 133 | 149 | 12,581 |
| Wsd-05 | 831 | 1701 | 7743 | 1936 | 139 | 302 | 120 | 12,772 |
| Wsd-06 | 630 | 456 | 10,526 | 817 | 41 | 384 | 66 | 12,919 |
| Wsd-07 | 904 | 3811 | 6209 | 1733 | 154 | 428 | 130 | 13,370 |
| Wsd-08 | 246 | 1520 | 14,725 | 1059 | 190 | 374 | 75 | 18,189 |
| Wsd-09 | 560 | 2919 | 12,907 | 1900 | 183 | 507 | 190 | 19,166 |
| Wsd-10 | 888 | 4278 | 13,011 | 2573 | 330 | 319 | 258 | 21,658 |
| Wsd-11 | 1377 | 9267 | 19,252 | 4141 | 479 | 846 | 414 | 35,775 |
| Wsd-12 | 1393 | 6160 | 29,910 | 2967 | 489 | 755 | 573 | 42,246 |
| Wsd-13 | 2835 | 5424 | 29,165 | 2911 | 548 | 1033 | 843 | 42,759 |
| Wsd-14 | 2552 | 8745 | 28,863 | 3615 | 716 | 881 | 596 | 45,968 |
| Wsd-15 | 1566 | 6316 | 65,558 | 4638 | 663 | 1750 | 615 | 81,107 |

**Table 3.** Statistics of monthly streamflow and rain gauge station data.

| Watershed | Monthly Streamflow ($\times 10^6$ m³) | | | Number of Used Rain Gauge Stations |
|---|---|---|---|---|
| | Minimum | Maximum | Mean | |
| Wsd-01 | 0.14 | 30.38 | 3.44 | 2 |
| Wsd-02 | 1.05 | 44.82 | 5.12 | 1 |
| Wsd-03 | 0.003 | 69.72 | 7.39 | 1 |
| Wsd-04 | 0.14 | 79.58 | 9.27 | 3 |
| Wsd-05 | 0.18 | 57.88 | 6.52 | 2 |
| Wsd-06 | 0.42 | 41.99 | 8.42 | 2 |
| Wsd-07 | 0.22 | 214.75 | 11.32 | 2 |
| Wsd-08 | 0.06 | 121.00 | 8.14 | 1 |
| Wsd-09 | 0.10 | 147.04 | 14.11 | 4 |
| Wsd-10 | 0.17 | 104.68 | 10.73 | 3 |
| Wsd-11 | 0.90 | 151.18 | 20.41 | 1 |
| Wsd-12 | 0.13 | 199.93 | 23.94 | 2 |
| Wsd-13 | 3.71 | 158.96 | 26.86 | 3 |
| Wsd-14 | 0.21 | 177.73 | 17.78 | 1 |
| Wsd-15 | 7.38 | 462.88 | 46.50 | 1 |

## 3. Results

### 3.1. Model Calibration

The measured values of the streamflow were compared to estimated values of the hydraulic model to determine whether the estimated values match the measured values. Duda et al. [27] claimed that the values are adequate when the difference is 45% or below and $R^2$ is above 0.65, whereas Skaggs et al. [28] claimed that they are adequate when NSE is greater than 0.5. Additionally, the criteria for adequate estimated values according to Wang et al. [29] were that $R^2$ should be above 0.60, NSE above 0.50, and PBIAS $\pm15\%$, whereas Moriasi et al. [30] declared that they accepted NSE above 0.50, $R^2$ above 0.60, and PBIAS below 15%. Thus, although criteria including NSE are used to determine the adequacy of estimated values, there are various other criteria that determine adequacy. The adequacy of estimated values from the L-THIA 2022 in this study was determined with NSE or $R^2$, based on scatter plots.

The L-THIA 2022 model was calibrated by comparing measured streamflow from January 2011 to December 2015 for all watersheds. The adjustment for CNs was determined by land use and HSGs in each watershed, and coefficients for each land use, were calculated using Equation (7), based on the L-THIA 2022 interface (Figure 2c,d). NSE ranged from 0.601 (Wsd-07) to 0.868 (Wsd-03) and $R^2$ from 0.743 (Wsd-15) to 0.917 (Wsd-03) (Table 4). The scatter plots were mostly expressed linearly from the smallest to greatest values of the estimated and measured values in each watershed (Figure 4).

### 3.2. Model Validation

Validation of the L-THIA 2022 model was performed by comparing the estimated and measured streamflows from January 2016 to December 2020 in all 15 watersheds. NSE ranged from 0.611 (Wsd-04) to 0.917 (Wsd-02) and $R^2$ from 0.676 (Wsd-15) to 0.946 (Wsd-02) (Table 5). Similar to the calibration process, the estimated streamflow from L-THIA 2022 was ascertained to be reliable during the validation process. NSE was 0.7 or above for all watersheds except for Wsd-04, Wsd-07, and Wsd-15, whereas $R^2$ was 0.7 or above for all watersheds except for Wsd-15. Similar to the scatter plots from calibration, the scatter plots from the validation process were expressed linearly going from the smallest to the greatest values of both the estimated and measured values for each watershed (Figure 5).

**Table 4.** NSE and R$^2$ values of measured and estimated monthly streamflows during calibration.

| Watershed | NSE | R$^2$ |
|---|---|---|
| Wsd-01 | 0.834 | 0.867 |
| Wsd-02 | 0.729 | 0.873 |
| Wsd-03 | 0.868 | 0.917 |
| Wsd-04 | 0.755 | 0.865 |
| Wsd-05 | 0.766 | 0.795 |
| Wsd-06 | 0.813 | 0.863 |
| Wsd-07 | 0.601 | 0.765 |
| Wsd-08 | 0.739 | 0.882 |
| Wsd-09 | 0.799 | 0.879 |
| Wsd-10 | 0.866 | 0.867 |
| Wsd-11 | 0.776 | 0.861 |
| Wsd-12 | 0.854 | 0.874 |
| Wsd-13 | 0.770 | 0.875 |
| Wsd-14 | 0.776 | 0.808 |
| Wsd-15 | 0.713 | 0.743 |

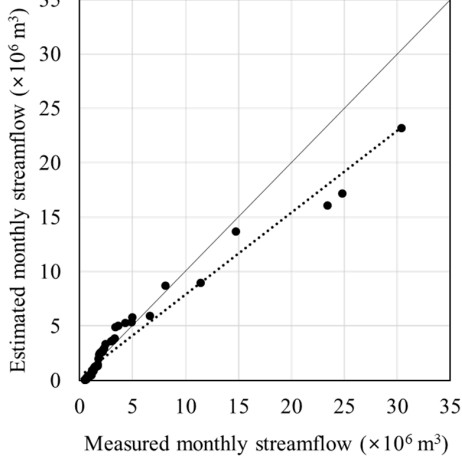

(**a**)

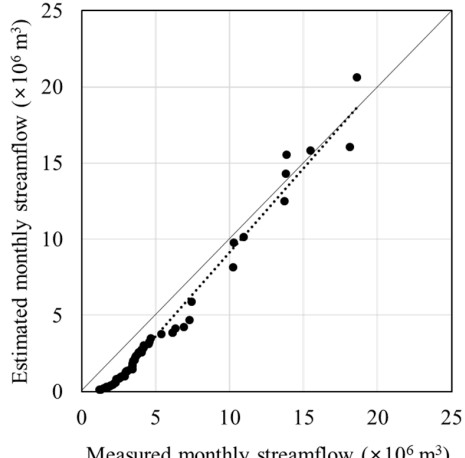

(**b**)

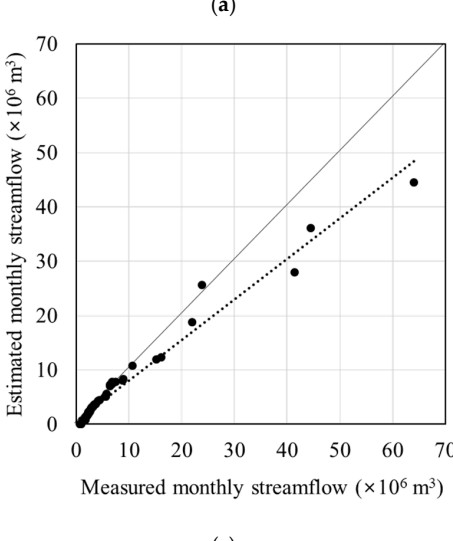

(**c**)

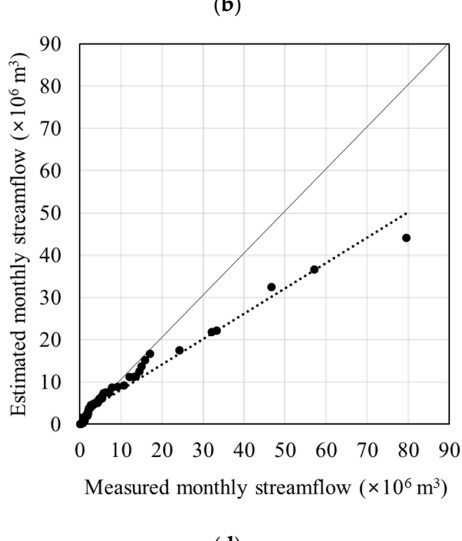

(**d**)

**Figure 4.** *Cont.*

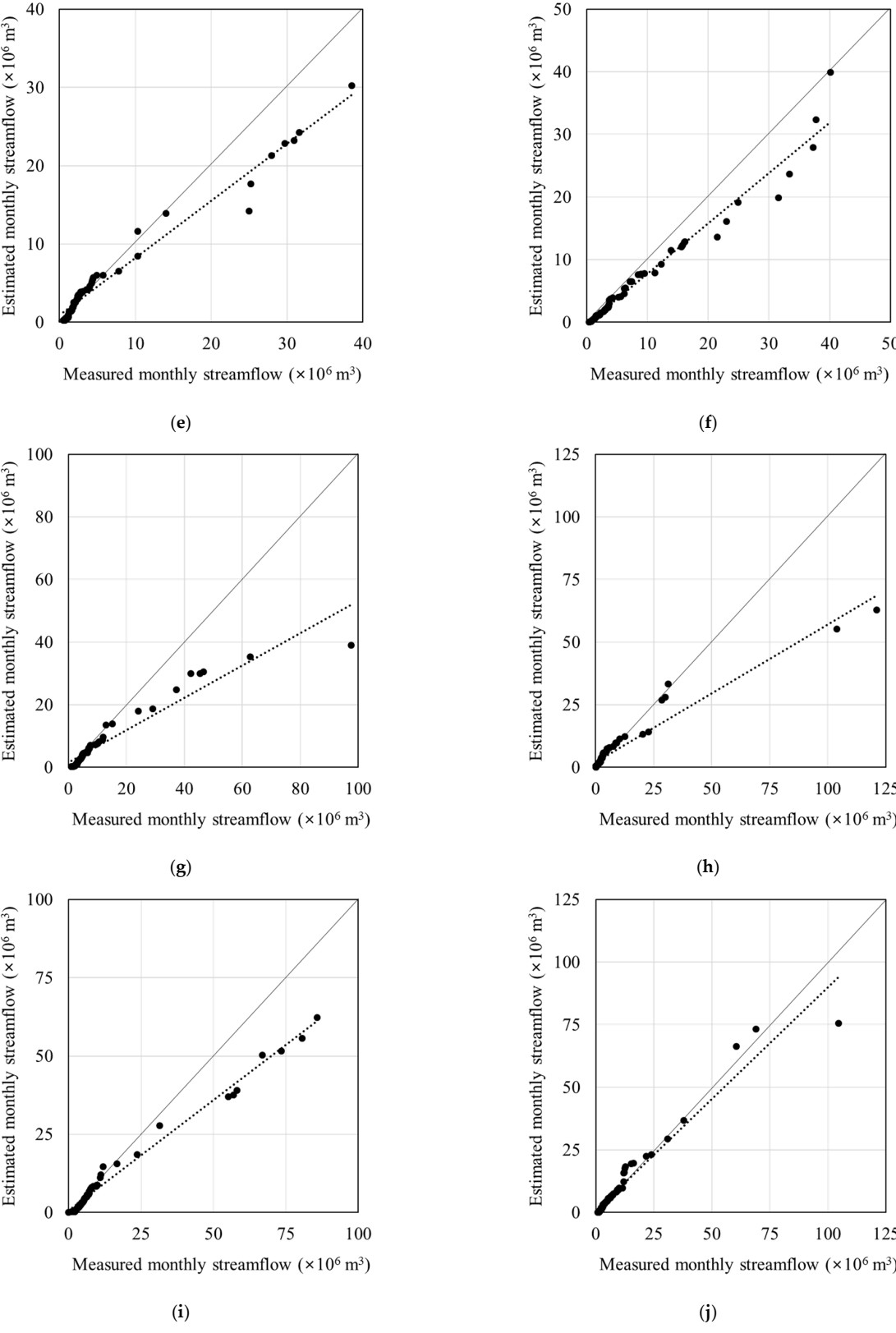

**Figure 4.** *Cont.*

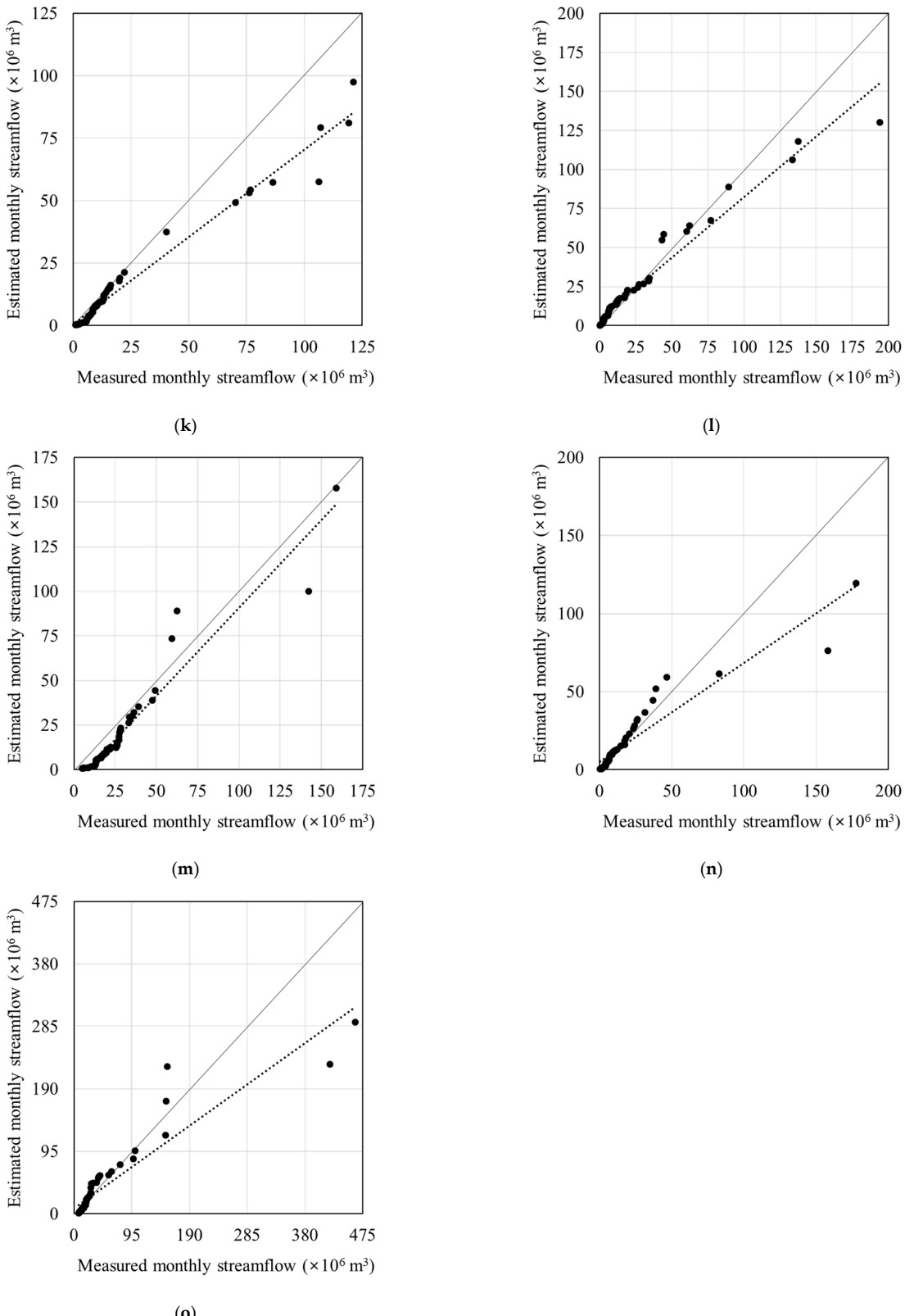

**Figure 4.** Scatter plots of measured and estimated monthly streamflows during calibration: (**a**) Wsd-01; (**b**) Wsd-02; (**c**) Wsd-03; (**d**) Wsd-04; (**e**) Wsd-05; (**f**) Wsd-06; (**g**) Wsd-07; (**h**) Wsd-08; (**i**) Wsd-09; (**j**) Wsd-10; (**k**) Wsd-11; (**l**) Wsd-12; (**m**) Wsd-13; (**n**) Wsd-14; and (**o**) Wsd-15.

**Table 5.** NSE and R$^2$ of measured and estimated monthly streamflows during validation.

| Watershed | NSE | R$^2$ |
|---|---|---|
| Wsd-01 | 0.785 | 0.808 |
| Wsd-02 | 0.917 | 0.946 |
| Wsd-03 | 0.746 | 0.878 |
| Wsd-04 | 0.611 | 0.726 |
| Wsd-05 | 0.780 | 0.805 |
| Wsd-06 | 0.753 | 0.817 |
| Wsd-07 | 0.641 | 0.823 |
| Wsd-08 | 0.687 | 0.700 |
| Wsd-09 | 0.714 | 0.834 |
| Wsd-10 | 0.719 | 0.729 |
| Wsd-11 | 0.900 | 0.913 |
| Wsd-12 | 0.730 | 0.797 |
| Wsd-13 | 0.718 | 0.754 |
| Wsd-14 | 0.865 | 0.868 |
| Wsd-15 | 0.636 | 0.676 |

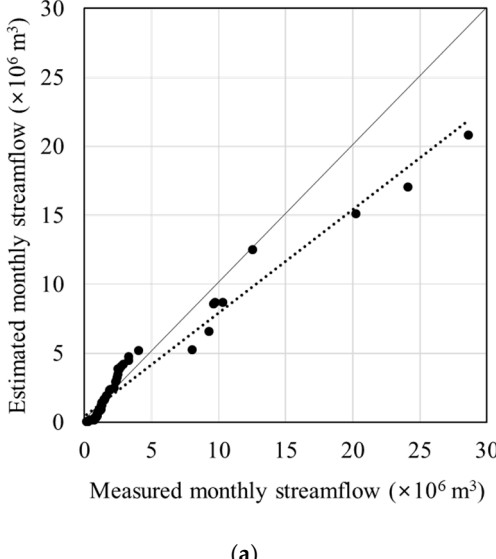

(**a**)

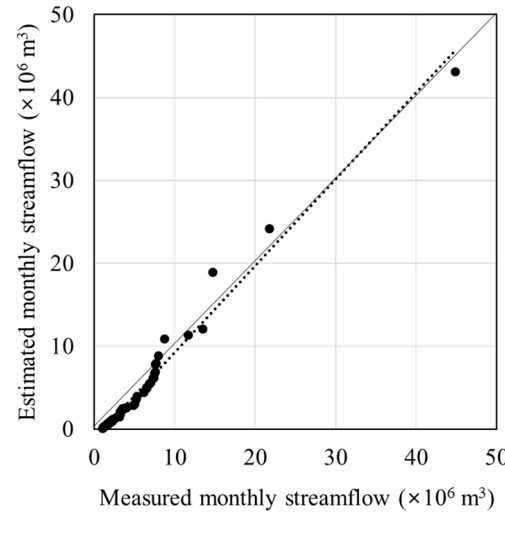

(**b**)

**Figure 5.** *Cont.*

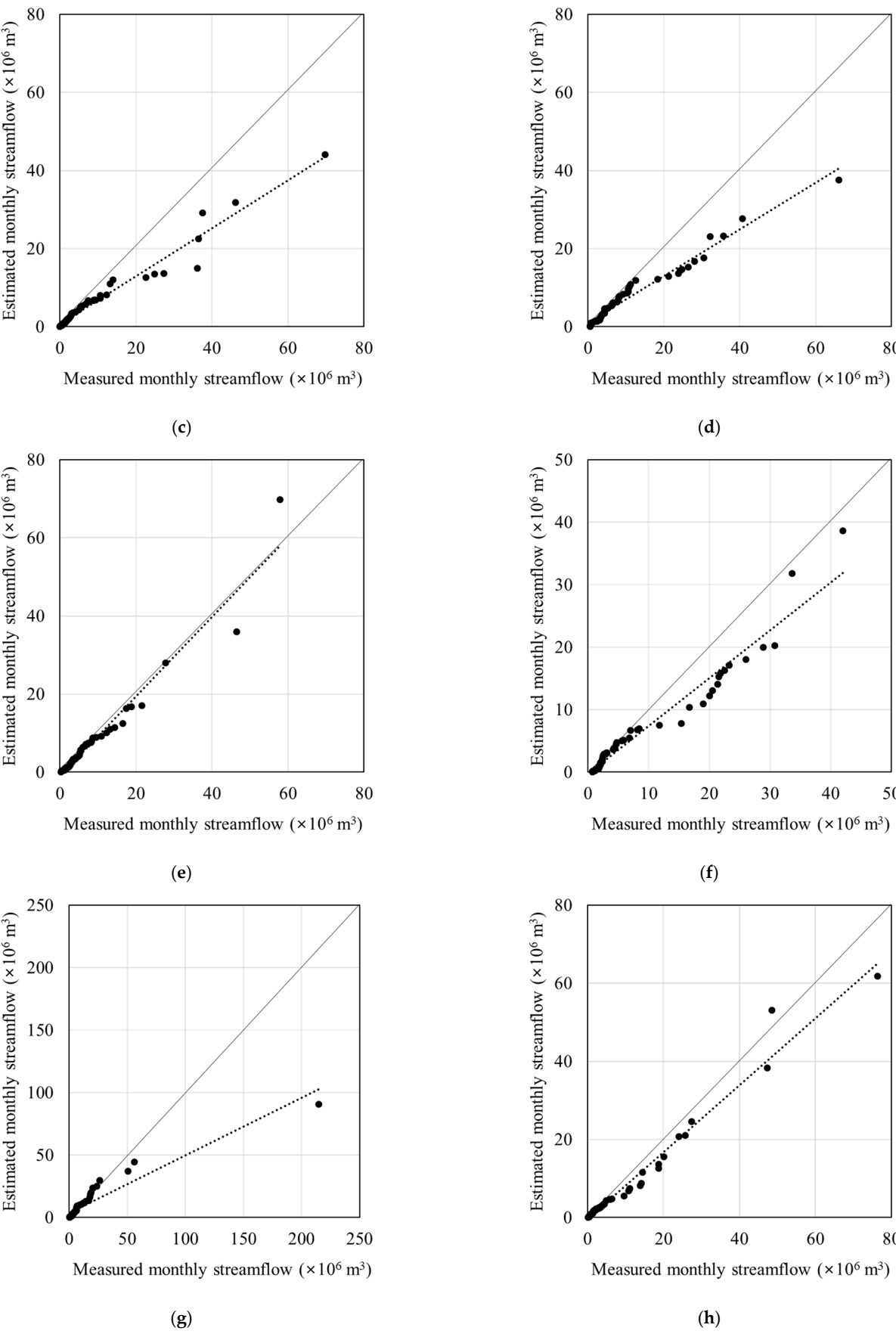

**Figure 5.** *Cont.*

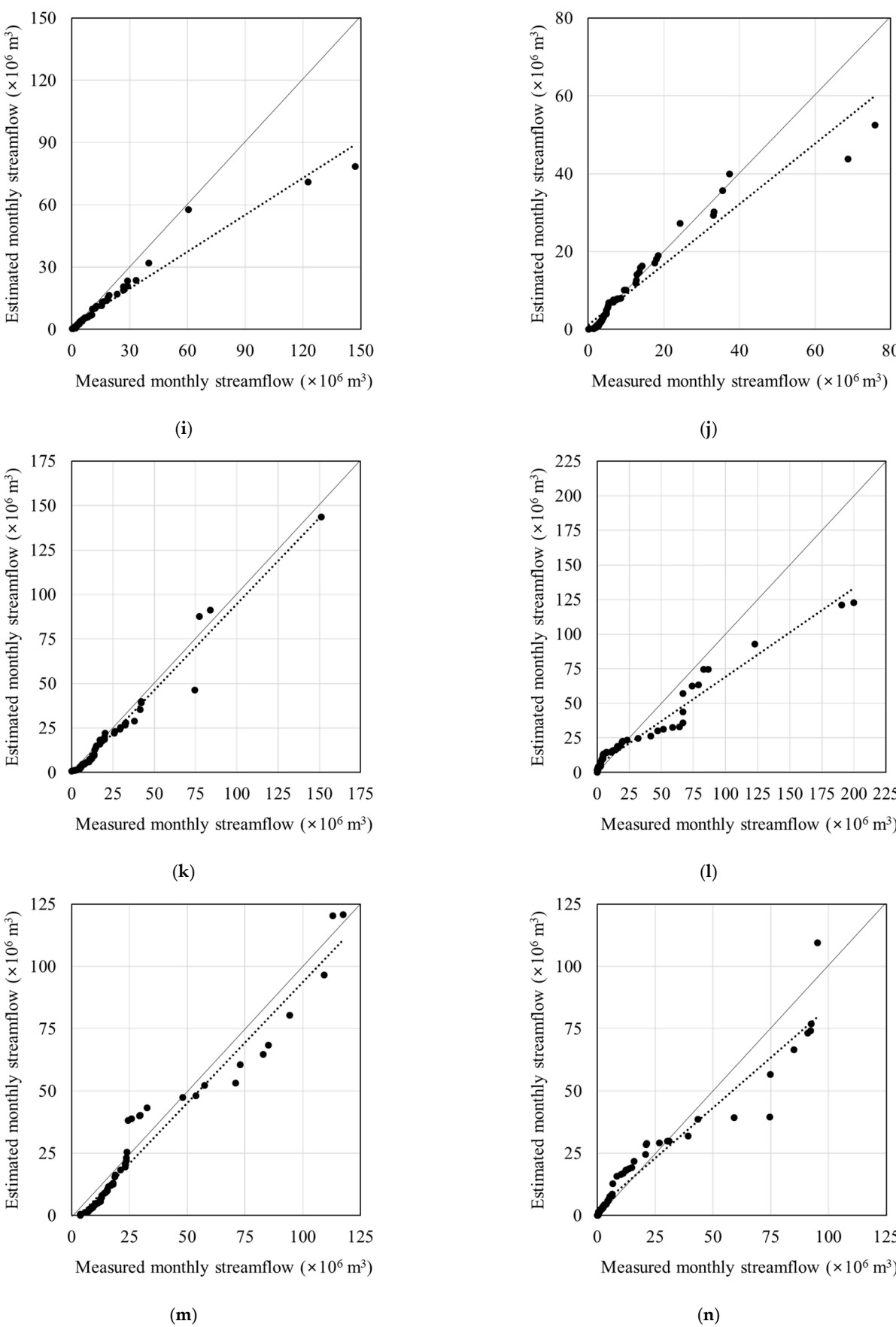

**Figure 5.** *Cont.*

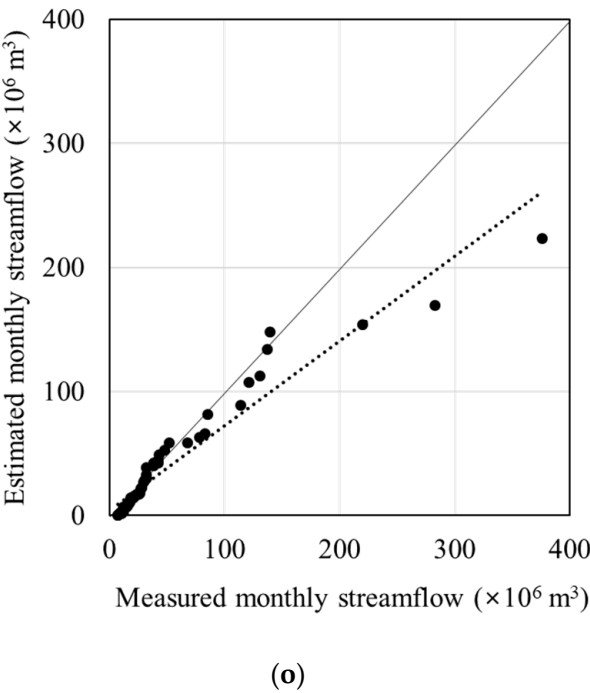

(**o**)

**Figure 5.** Scatter plots of measured and estimated monthly streamflows during validation: (**a**) Wsd-01; (**b**) Wsd-02; (**c**) Wsd-03; (**d**) Wsd-04; (**e**) Wsd-05; (**f**) Wsd-06; (**g**) Wsd-07; (**h**) Wsd-08; (**i**) Wsd-09; (**j**) Wsd-10; (**k**) Wsd-11; (**l**) Wsd-12; (**m**) Wsd-13; (**n**) Wsd-14; and (**o**) Wsd-15.

## 4. Conclusions

Variations in rainfall pattern or land use can lead to changes in the hydraulic behavior of watersheds, for which spatial and temporal characteristics must be considered through land use maps that reflect ground surface conditions; for example, an HSG map reflects the conditions such as permeability of the subsurface soil and the rainfall data from numerous locations. These conditions are defined as numerous HRUs by the combination of land use and spatial locations of HSG. Rainfall data measured at points adjacent to the subwatershed need to be applied differently for each HRU, making the use of GIS software necessary for watershed management.

This study developed an L-THIA 2022 model that is operable alongside the QGIS software. This model defines HRU according to the spatial distributions in land use and HSG maps, in which the spatial locations of rainfall data measured from numerous points can be properly considered. In general, estimations for streamflow are performed more often than those for direct runoff. As this model was improved to enable streamflow estimation, the immediate comparison of measured and estimated values became possible. This is likely to considerably enhance the applicability of the L-THIA model by overcoming the previously impossible streamflow estimation, due to the absence of baseflow estimation from the existing L-THIA model. The L-THIA 2022 model was applied to 15 watersheds with different areas to determine the applicability of the model. In the calibration process, NSE was 0.7 or above for all watersheds excluding one watershed, and $R^2$ was 0.7 or above for all watersheds. Therefore, model calibration was considered successful, and estimated streamflow were ascertained for their reliability. In addition, NSE was 0.7 or above for all watersheds except three out of fifteen watersheds, $R^2$ was 0.7 or above for all watersheds except only one watershed in the validation process. Moreover, the scatter plots were expressed linearly going from the smallest to the greatest values of both the estimated and measured values for each watershed in both calibration and validation processes. Therefore, results indicated that the model was reliable during both the calibration and validation processes.

The L-THIA 2022 model, operated along with QGIS, is able to consider the spatial distribution of land use and use rainfall data measured from numerous points, making it useful for analyzing the land use changes such as urbanization, as well as rainfall pattern changes caused by factors such as climate change.

**Author Contributions:** Conceptualization, Youn Shik Park; methodology, Jong-Yoon Park; investigation, Jong-Yoon Park; data curation, Min Suh Chae; writing—original draft preparation, Hanyong Lee; writing—review and editing, Kyoung Jae Lim; visualization, Jong-Yoon Park; supervision, Youn Shik Park. All authors have read and agreed to the published version of the manuscript.

**Funding:** This subject is supported by the Korea Ministry of Environment as "The SS (Surface Soil conservation and management) projects; 2019~(2019002820001)".

**Institutional Review Board Statement:** Not applicable.

**Informed Consent Statement:** Not applicable.

**Data Availability Statement:** The data presented and used in the study are available on the request from the corresponding author.

**Acknowledgments:** The authors send special thanks to the Korea Environmental Industry & Technology Institute for continuous research project support.

**Conflicts of Interest:** The authors declare no conflict of interest.

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
