# Peer review of "Development and Application of a QGIS-Based Model to Estimate Monthly Streamflow"

_ijgi, doi:10.3390/ijgi11010040_

Round 1

Reviewer 1 Report

The paper deals with a very important topic of streamflow estimation / modelling. It showcases a novel approach and provides significant results of testing such approach in actual real-world conditions. The overall level of English and clarity is good, the methodology and scientific approach used is appropriate. However, I do believe that there are two areas where the paper should be improved prior to publishing:

  1. In my opinion the authors do not sufficiently address the question "why" they are developing this model. Models that estimate streamflow already exist and are widely in use all over the world (for example SWAT model). The paper does not sufficiently justify the need for a new model to be developed. It would benefit the paper if authors specified what are the issues / problems with existing models that they were trying to address with their novel approach. What are the main differences between these models?
  2. The methodology used for assigning gauging points to watersheds is unclear and there are some statements in the paper that seem to contradict each other. The table on line 237 shows different numbers of gauging stations being used for each watershed, which alongside the statement on line 167: "based on rainfall data from multiple gauging points" leads me to believe that some sort of distance-weighted calculation was performed. However the sentence on line 163 clearly states that: "to apply rainfall data from the nearest rain gauge", which is in direct contradiction, because that would mean only one gauge was used for used for each watershed. Either this is just an error in writing (perhaps due to a language barrier), or the discrepancy of using two distinct approaches in different parts of the model has to be explained.

There are also some minor formal mistakes, such are mislabeled tables (there are two "table 2" in the paper - one on line 225 and one on 237). The initial abstraction Iis supposed to be capital "I", but in the text on lines 82 and 93, there is clearly small letter "L" used instead.

Author Response

Dear the reviewer,

Thank you for your valuable comments.

Thank you.

Reviewer 2 Report

This manuscript discussed a method using QGIS to estimate streamflow in 15 watersheds based on: land use, soil group, and rainfall data. I do not find this manuscript a good fit for publication because it is significantly underdeveloped. Below are some detailed comments regarding the decision.

Comments

  1. The manuscript does not read like a scientific paper at all. Some part of it looks straight out of a hydrology book, and the rest seems like a simple technical report.
  2. The contribution of this study is unclear. There are tons of complex work doing streamflow estimation based on the land use pattern, soil type, and rainfall data. This study does not look significant in that sense. It should be highlighted what the major knowledge gaps are and how this study addresses them.
  3. Line 93-94: Should be Ia, instead of 1a.
  4. Adding an identity line to the plots would help to visualize the deviations clearly.
  5. The results section only shows some figures and tables and reports those numbers in the paragraph. There is no insightful decision about the results.

Author Response

(The authors gave the same response as above.)

Reviewer 3 Report

The paper is well written and the topics are adequately and clearly covered.

I recommend some formal/minor corrections:

1. Line 180, there is a typo in the model name. The model name is L-THIA 2022 not 2002, right?
2. In general, reduce white space in the text and tables.
3. Separate results from discussions in section "3. Results and discussion". Discussions could be added to the conclusions.
4. Line 314, the link currently does not work. Put supplementary materials in an appendix, if possible.

Author Response

(The authors gave the same response as above.)

Reviewer 4 Report

Accepted with minor revisions

A novel work, that merits publication with minor revisions.

It shows how changes in rainfall pattern and land use have caused considerable impacts on the hydrological behavior of watersheds, using the  Long-Term Hydrologic Impact Analysis (L-THIA) model. The fact that this model is tested in 15 watersheds, is a good beginning for the globalization of the model in other areas, as well and watershed's sizes. 

Some reference enrichment, from areas all over the world, would add some more value to your work, such as : 

Thakur P.K. et al. (2022) Role of Geospatial Technology in Hydrological and Hydrodynamic Modeling-With Focus on Floods Studies. In: Pandey A., Chowdary V.M., Behera M.D., Singh V.P. (eds) Geospatial Technologies for Land and Water Resources Management. Water Science and Technology Library, Vol 103. Springer, Cham. https://doi.org/10.1007/978-3-030-90479-1_26

Sentas A., Karamoutsou L., Charizopoulos N., Psilovikos T., Psilovikos A. & Loukas A. (2018). The use of Stochastic Models for Short Term Prediction of Water Parameters of the Thesaurus Dam, Nestos River, Greece. 3rd EWaS International Conference 2018, Vol 2 (634), pp 1-6, Lefkada, Greece.

Pochaievets O., Obodovskyi O., Lukianets O. & Grebin V. (2021). Algorithm research and evaluation of minimum water flow of mountain rivers using GIS. EAGE, Geoinformatics 2021, 11-14 May 2021, Online Event.

Author Response

(The authors gave the same response as above.)

Round 2

Reviewer 2 Report

N/A